# A High Prevalence of Cardiopulmonary Worms Detected in the Iberian Wolf (*Canis lupus*): A Threat for Wild and Domestic Canids

**DOI:** 10.3390/ani12172289

**Published:** 2022-09-03

**Authors:** Efrén Estévez-Sánchez, Rocío Checa, Ana Montoya, Juan Pedro Barrera, Ana María López-Beceiro, Luis Eusebio Fidalgo, Guadalupe Miró

**Affiliations:** 1Departamento de Sanidad Animal, Facultad de Veterinaria, Universidad Complutense de Madrid, Avda. Puerta de Hierro s/n, 28040 Madrid, Spain; 2Departamento de Anatomía, Producción Animal y Ciencias Clínicas Veterinarias, Facultad de Veterinaria, Universidad de Santiago de Compostela, Campus Terra, 27002 Lugo, Spain

**Keywords:** *Angiostrongylus vasorum*, *Eucoleus aerophilus*, *Crenosoma vulpis*, *Canis lupus signatus*, northwestern Spain

## Abstract

**Simple Summary:**

The Iberian wolf (*Canis lupus signatus*) is a recognized reservoir of some zoonotic parasites that cause diseases transmissible to domestic animals and/or humans. The objective of this study was to determine the diversity of species of cardiopulmonary nematode parasites that affect wolves in northwestern Spain, and to estimate their prevalence and the relationship between these parasites and various epidemiological variables. The cardiopulmonary systems of 57 wolves from Galicia were examined using dissection and cup sedimentation techniques, and the collected worms were then morphologically identified. The overall prevalence of infection by cardiopulmonary nematodes was 24.5%, and the parasite species identified were *Angiostrongylus vasorum* (19.3%), *Crenosoma vulpis* (7%) and *Eucoleus aerophilus* (3.5%). The latter is of zoonotic interest. A significant relationship was found between age and *C. vulpis* infection, which was only found in animals under one year of age. Our findings indicate that the Iberian wolf could play an important role in maintaining cardiopulmonary parasites in the wild, and they highlight a need to identify potential risks for veterinary and public health.

**Abstract:**

Cardiopulmonary nematodes are highly pathogenic parasites affecting domestic and wild canids. As the result of conservation programs, the Iberian wolf (*Canis lupus signatus*) population has recently expanded, and its distribution range covers lands from where it had long disappeared. However, the exact epidemiological role of the wolf in the life cycle of zoonotic parasites causing diseases transmissible to pets and/or humans is largely unknown. This study sought to determine the diversity of cardiopulmonary nematode parasite species that affect wolves inhabiting northwestern areas of the Iberian Peninsula, and to estimate their prevalence and the relationship between these parasites and several epidemiological variables. For this purpose, we examined the cardiopulmonary systems of 57 wolves from Galicia (from the provinces A Coruña *n* = 15, Lugo *n* = 21, Ourense *n* =15 and Pontevedra *n* = 6) using techniques of dissection and cup sedimentation. Collected worms were then identified under a light microscope according to their morphological features. Three species of nematodes were detected: *Angiostrongylus vasorum* (the “French-heartworm”), *Crenosoma vulpis* and *Eucoleus aerophilus*, the latter being of zoonotic interest. The prevalence was 24.5% (14/57; 95% CI 13.3–35.6%) overall, 19.3% for *A. vasorum* (11/57; 95% CI 8.8–29.2%), 7% for *C. vulpis* (4/57; 95% CI 0.4–13.6%) and 3.5% for *E. aerophilus* (2/57; CI −1.1–9.1%). A significant relationship (*p* = 0.002) was found between age and the presence of *C. vulpis*, which was only found in juvenile animals. Furthermore, a higher prevalence of *A. vasorum* and/or *C. vulpis* was observed in wolves with a lower body condition score (40% and 20%, respectively), though the difference was not significant (*p* = 0.221 and *p* = 0.444, respectively). Our findings indicate a high “French-heartworm” and lungworm burden in the wolf population of northern Spain, and they identify a need for studies designed to elucidate the epidemiological role played by the Iberian wolf and to identify possible risks for veterinary and public health.

## 1. Introduction

The Iberian wolf (*Canis lupus signatus*) is the second largest predator in the Iberian Peninsula, only surpassed by the Eurasian brown bear (*Ursus arctos arctos*). With an estimated population of 1700 to 2500 wolves in Spain [1], many inhabit its northwestern corner, comprising an area of 120,000 km^2^, mainly spanning the autonomous communities of Galicia, Asturias and Castilla-Leon [2]. The habitats of these carnivores are often anthropic rural areas [3] where they can feed on both wild and domestic ungulates. This added to the recent expansion of the wolf population across the Iberian Peninsula has rekindled the old debate about what should be done about these feared hunters.

Wolves play an important ecological and health role in the natural environment. They are predators of sick animals, preventing the transmission of diseases such as tuberculosis [4], or promoting vegetation regrowth by reducing the ungulate population [5]. However, they are also an important component of the life cycle of many emerging parasites, some of which are highly pathogenic and potentially lethal for companion animals and humans.

Parasitic diseases take on a special interest because of their possible impact on public health, but also for the conservation and protection of their host species [6]. After many years of recovery, wolf populations have expanded to inhabit areas throughout the Iberian Peninsula [7]. However, because of their wildlife style, it is difficult to preserve their health state and prevent diseases that threaten their conservation. Cardiopulmonary nematode worms cause chronic, usually subclinical diseases in domestic animals, but wolves require their full lung and heart capacity for daily life [8]. Therefore, these diseases can be fatal, especially in young animals.

Owing to their emergent nature and spread to thus far non-endemic places, some cardiopulmonary parasites, such as the metastrongyloids *Angiostrongylus vasorum* and *Crenosoma vulpis*, the tricurid *Eucoleus aerophilus* (syn. *Capillaria aerophila*), the spirurid *Dirofilaria immitis* and the family Filaroididae (*Oslerus osleri* and *Filaroides hirthi*), are starting to attract the interest of scientists. The exact cause of the increase in parasite numbers is unknown, but global warming, changes in vector seasonal population dynamics and movements in animal populations are thought to play an important role in this expansion [9]. 

*Angiostrongylus vasorum* (“French-heartworm”) and *Crenosoma vulpis* (lungworm) have an indirect biological cycle and affect wild carnivores such as the pine marten (*Martes martes*), red fox (*Vulpes vulpes*) and wolf (*Canis lupus*) [10,11,12]. The intermediate host is a land or aquatic gastropod [13] that spreads third-stage larvae (L3) by ingestion. The life of these gastropods is dependent on climate conditions of humidity and temperature, and they hibernate or aestivate (depending on the species) when these conditions become too harsh [14].

The lungworm *Eucoleus aerophilus* is a parasite affecting dogs, cats, wild carnivores and, sometimes, humans worldwide [15]. Its life cycle is not known exactly, but it seems to be transmitted directly by the fecal–oral route or through paratenic hosts (earthworms) harboring L1 larvae [16].

*Dirofilaria immitis* is a zoonotic parasite that mainly affects dogs and cats, although it can infest several mammals. Dirofilariosis is transmitted by mosquitoes (Diptera, Culicidae) and is endemic in southern Mediterranean countries including Spain [17]. 

The lungworms *Oslerus osleri* and *Filaroides hirthi* both affect canids. Those of the family Filaroididae are normally parasites of immunocompromised, stressed or young dogs, and only few cases have been reported in domestic animals in Spain [18,19,20]. Transmission from mothers to their puppies occurs via licking or through coprophagy [21].

Despite being such a relevant topic, few studies have addressed the epidemiology and pathogenesis of cardiopulmonary parasites in wolves in Europe. Spain is currently considered an endemic area, and the red fox (*Vulpes vulpes*) seems to be the main reservoir of these parasites [8,10,12]. Although their prevalence in Spanish domestic dogs is apparently unrepresentative (0.75–1.73%) [22,23], the life cycle of cardiopulmonary parasites in wild hosts has been established, affecting animals such as wolves, foxes and other mesocarnivores. As a result of this close relationship between the domestic and wild life cycle, these reservoirs could play a key role in parasite spread to pets and humans [24].

The aim of this study was to examine the presence of cardiopulmonary parasites in wolf populations of the northwestern Iberian Peninsula and to analyze the different epidemiological variables involved.

## 2. Materials and Methods

### 2.1. Study Area 

The study population comprised 57 Iberian wolves (*Canis lupus signatus*) from northwest Spain. These wolves died of natural causes or were killed in road accidents in the Galicia region, and they were collected by official organisms of Galicia over the period of 2016 to 2021. In this geographical area, climate is predominantly oceanic, with annual temperature averages of around 12.0–14.9 °C and rainfall averages of around 811–1791 mm in 2021 [25]. 

### 2.2. Sample and Data Collection

The following data were collected for each animal: age, sex, weight, body condition score and any other relevant findings in a physical examination. Age was classified according to body weight, dentition and sexual maturity, as: wolf pups (under 12 months of age and without full body development), immature or young (1 to 3 years of age and not yet sexually mature) and adults (over 3 years, or of reproductive age). Body condition was scored from 1 to 5, where 1 indicates extreme thinness, 3 indicates an optimal state and 5 is markedly overweight.

Wolves were necropsied following standard procedures. Cardiorespiratory tissue samples were obtained by sectioning the neck muscles and dissecting adjacent tissues to harvest the laryngeal cartilages, trachea, lungs and heart in a single block. Biological samples were stored frozen (−20 °C) until their laboratory analysis.

### 2.3. Cardiorespiratory Parasites 

The trachea, bronchi and bronchioles were longitudinally dissected and inspected to collect as many nematodes as possible. The heart chambers and blood vessels were also carefully dissected following the blood circulation. Once the lungs and heart were fully dissected, they were deposited in a tapered graduated sedimentation cup. The cups were filled with warm water until the organs were completely submerged and set to rest for at least 6 h to promote worm sedimentation. The sediment was filtered, and the remaining material was examined under a stereomicroscope to identify the worms [26,27].

All worms collected during this procedure were deposited in tubes containing 70% ethanol for less than 72 h. Subsequently, the parasites were deposited in a glass dish and identified morphologically under a light microscope. Worms were classified at the species level according to sex (male or female) using identification keys [16,28,29,30]. The total number of parasites collected from each cardiorespiratory system was also recorded. 

### 2.4. Statistical Analysis

The prevalence (percentage of infected animals), parasite intensity (number of parasites *per* infected animal) and species richness (number of different species parasitizing the same host) were reported. The differences observed according to epidemiological variables (age, sex, body condition and province) and the significance of each association (between prevalence and each epidemiological variable) were analyzed using Fisher’s exact test at a 95% confidence interval (significance was set at *p* ≤ 0.05). The software employed for this analysis was R version 4.2.1.

## 3. Results

### 3.1. Nematode Parasite Species and Prevalence

The overall proportion of wolves detected positive was 24.5% (14/57; 95% CI 13.3–35.6%). The species identified were *Angiostrongylus vasorum*, *Crenosoma vulpis* and *Eucoleus aerophilus*. Other species such as *Dirofilaria immitis*, *Oslerus osleri* and *Filaroides hirthi* were not found by necropsy in these animals. 

The individual prevalence was: *A. vasorum* 19.3% (11/57; 95% CI 8.8–29.2%), *C. vulpis* 7% (4/57; 95% CI 0.4–13.6%) and *E. aerophilus* 3.5% (2/57; CI −1.1%–9.1%). Eleven (19.3%) of the wolves were infected by one species, and three (5.2%) had a two-species coinfection (Table 1). No animal was infected by the three different species.

The total number of worms collected was 385, with 275 females (71.4%) and 72 males (18.7%). Sex could not be determined in 38 worms. The number of parasites found in the post mortem exam of a single infected wolf ranged from 1 to 99 parasites (mean 27.5 ± 33.4). Furthermore, in the case of *A. vasorum*, the number of worms *per* animal (single infections and co-infections) ranged from 1 to 75 and from 1 to 24 for female and male worms, respectively. 

All wolves infected by *A. vasorum* had at least one female worm (11/11), and 82% had at least one male worm (9/11). In addition, 25% of the animals infected by *C. vulpis* had one male worm (1/4), and only female worms were identified in animals harboring *E. aerophilus* (Table 1).

### 3.2. Epidemiological Data Recorded in Wolves with Nematode Worms

Of the 57 Iberian wolves examined, 21 were from the Galician province of Lugo (37%), 15 were from A Coruña (26%), 15 were from Ourense (26%) and 6 were from Pontevedra (11%). A total of 29 were male (51%), and 28 were females (49%). A total of 14 were classified as wolf pups (25%), 24 were classified as young (24%) and 19 were classified as adults and older adult wolves (33%). For the whole population, body condition scores (out of 5) were: 22% 1–2 (*n* = 12), 35% 3 (*n* = 20) and 45% 4–5 (*n* = 24).

A higher overall worm prevalence was observed in the A Coruña and Ourense provinces (26.7%), followed by Lugo (23.8%) and Pontevedra (16.7%) (Figure 1). By species, the higher prevalence of *A. vasorum* was detected in Lugo (23.8%), whereas infection by *C. vulpes* was only observed in A Coruña and Ourense (13.3%) and by *E. aerophilus* in the provinces of Ourense and Lugo (6.7% and 4.8%, respectively) (Table 2). Notwithstanding, differences in cardiopulmonary worm prevalence between provinces were not significant (*p* = 1).

Nearly similar overall prevalence was detected in males and females, with 24.1% and 25%, respectively. However, infection by *A. vasorum*, *C. vulpis* and *E. aerophilus* was higher in females (21.4%, 7.1% and 7.1%, respectively), although these differences were not significant (Table 2).

Wolves with a lower body condition score showed a higher prevalence of *A. vasorum* and/or *C. vulpis* (40% and 20%, respectively), yet the difference was once again not significant. However, animal age and the presence of *C. vulpis* were significantly associated, such that this lungworm species was only found in juvenile animals (*p* = 0.002). 

To compare worm infection intensity according to host age, nematode numbers were grouped into four categories. *p*-values were calculated comparing an age group of wolves with the remaining wolves for each grade of parasitization (Table 3). Hence, all infected wolf pups (42.9%) had fewer than 30 parasites *per* animal, and among the infected young wolves and adults, 8.3% and 5.26% had fewer than 30 parasites, 8.3% and 5.26% had between 30 and 60 parasites and 4.2% and 5.26% had more than 60 parasites, respectively. A significant link was thus observed between a low number of parasites (less than 30) and a less than one-year wolf age (*p* = 0.004).

## 4. Discussion

All three cardiopulmonary nematodes identified in our study (*A. vasorum*, *C. vulpis* and *E. aerophilus*) have been previously described in wolves from the Iberian Peninsula [6,8,11,12,31] and from other European countries [16,32,33,34,35,36]. It should also be noted that *O. osleri*, *F. hirthi* and *D. immitis*, reported previously in wolves from northwestern Spain, were not found in the present study. For Europe, few studies have provided lungworm and heartworm prevalence data in wolves. *A. vasorum* has been described in wolves from Slovakia (0.8%) [32], Croatia (3.1%) [33] and Italy (28%). Compared to Spain, *C. vulpis* is more prevalent in wolves from countries such as Latvia (9.1%) [35], Portugal (9.1%) [6] and Belarus (7.7%). Similarly, *E. aerophilus* shows a high prevalence in Europe (9–36%) [6,32,33,35]. 

There are mainly four techniques used for the detection of cardiopulmonary parasites: Baermann migration–sedimentation, immunological diagnosis through antigen- or antibody-based assays, molecular detection (polymerase chain reaction) and dissection identification [37,38,39,40,41]. This determines that the reported prevalence data are difficult to interpret because of the different sensitivity and specificity of the diagnostic techniques used. For wild animals, feces samples collected directly from the ground are often used. However, first-stage *A. vasorum* and *C. vulpis* larvae (L1) are susceptible to environmental conditions, and these studies are likely underestimating the prevalence. In contrast, *Eucoleus aerophilus* releases very resistant eggs [9], so data for this nematode could be considered more representative. This makes it difficult to make any comparison of prevalence data between different countries or regions. Furthermore, this study has some potential limitations: first, not all wolves were suitable for necropsy because of the bad preservation state of the tissues, and the limited sample size could have affected the statistical analysis; second, no other diagnostic tests have been performed to compare the results due to the lack of samples.

Our prevalence data for *A. vasorum* (19.3%) were similar to the reported rates based also on anatomical dissection in wolves from northwestern Spain (2.1–22%) [8,12]. These rates are lower than the *A. vasorum* prevalence reported in red foxes (*Vulpes vulpes*) (16–43.2%) [8,42,43,44,45], similar to those described for the Eurasian badger (*Meles meles*) (6.4–24%) [45,46], and higher than those provided for dogs (0.73–1.73%) [22,23] in Spain based on serology. Dogs do not usually feed on intermediate hosts [22], and wolves prefer larger prey [13]. However, terrestrial gastropods (intermediate hosts), anurans and birds (paratenic hosts) are a common food source for the fox and badger, and this is an accepted explanation for the different prevalence found among species [45].

In foxes, *A. vasorum* infection does not produce an efficient immune response, so adults previously exposed to this parasite can become reinfected and even sustain a persistent infection [47]. Although no study has been carried out in wolves, the similar prevalence detected here among juveniles and adults (21.7% and 23.3%, respectively) could mean that wolves also have no efficient immune response against this parasite. To better understand this, we should consider that wolves younger than one year showed an overall higher prevalence of cardiopulmonary parasites (42.9%) compared to young wolves (between 1 and 3 years) (28.8%), and in the specific case of *A. vasorum*, a similar prevalence was detected at both ages (21.4% and 20.8%, respectively). This means that only in the case of *A. vasorum* is there a stable (slightly upward) prevalence trend with animal age. In addition, young and adult animals showed a greater extent of parasitization with *A. vasorum*, in that 13.56% showed moderate infection, and 9.46% showed severe infection, which means that they could have persistent infections and/or be continuously reinfected.

Angiostrongylosis is an emerging disease causing severe respiratory and central nervous system signs in dogs that has spread throughout central and northwestern Europe. It is today considered endemic in many countries. Recent reports demonstrate that canine *A. vasorum* is expanding and that new *foci* are appearing in previously non-endemic areas [48]. Although the red fox is thought to be responsible for this expansion, the wolf may also play an important epidemiological role as a reservoir in the wildlife cycle, infecting other wild or domestic animals. In the last two decades, an increase in the prevalence of *A. vasorum* infection has been reported in the wolf population of the northwestern Iberian Peninsula from 2% [12] to 22–24.5% [8]. Spain is currently considered a stable *focus* of a high prevalence of *A. vasorum* due to its conditions of temperature and moisture, which facilitate the survival of intermediate hosts and first-stage larvae (L1) in the environment [49].

In Spain, *Crenosoma vulpis* has been described in the wolf, red fox and dog. The prevalence in wolves detected here (7%) was slightly lower than in previous studies carried out in the northwestern Iberian Peninsula (9.1–9.4%) [6,8]. The red fox is the main reservoir of *C. vulpis* in the Iberian Peninsula, and its prevalence is significantly higher than in the wolf (2.5–44.8%) [8,42,44,49]. To our knowledge, only one study has provided prevalence data for *C. vulpis* in Spanish dogs (2%) [50]. Our results reveal differences between *C. vulpis* and *A. vasorum* in wolves: (i) *C. vulpis* was only detected in juvenile wolves (28.6%), with the difference with older animals being significant (*p* = 0.002). This could be explained by the smaller prey caught by young wolves, but it could also reflect an efficient immune response against *C. vulpis* in adult wolves; and (ii) *C. vulpis* has been described in geographical areas lacking the presence of *A. vasorum* or where cases are only sporadic. Examples of these regions are Canada, Finland, Norway, Ireland or the United Kingdom [51], as first-stage *C. vulpis* larvae (L1) can remain active at low temperatures even after a freezing period of several days [52].

When infected wolves were classified according to body condition, those described as mildly thin were found to harbor the highest prevalence of parasites (50%). Thus, 40% of individuals with this body condition were parasitized with *A. vasorum*, 20% were parasitized with *C. vulpis* and none were parasitized with *E. aerophilus*. Thus, it seems that the most affected animals were infected by *A. vasorum* and *C. vulpis*, as these parasites, respectively, attack the vascular endothelium [11] and have an irritant effect on the respiratory system [12].

*Eucoleus aerophilus* has been described in both domestic and wild carnivores from Spain [31,43,46,53]. The prevalence of this nematode in wolves in northwestern Spain (4%) was lower than previously described (5.4–50.54%) [8,54]. In Spain, *E. aerophilus* is also more prevalent in the red fox (4.4–34.2%) [8,42,55,56] and pine marten (*Martes martes*) (50.98%) [10], but studies assessing this infection in domestic animals have been scarce. Unlike *A. vasorum* and *C. vulpis*, *E. aerophilus* plays a direct role in the life cycle for which the ingestion of eggs or paratenic hosts such as lumbricids are required [24]. However, its exact biological life cycle is unknown, so more studies are needed to understand the impact of these results.

The role of terrestrial gastropods and earthworms is of paramount importance in the distribution and prevalence of cardiopulmonary parasites. This relationship can be observed when comparing earthworm and mollusc distribution maps [57,58] with endemic areas of the parasites throughout Europe. It may be observed that Pontevedra has a lower abundance of earthworms and terrestrial gastropods, possibly explaining the lower prevalence of cardiopulmonary parasites detected in this province compared to the others included in the present study.

In conclusion, our observations indicate that the Iberian wolf could play an important role in the maintenance of cardiopulmonary parasites in the wild. Some studies have confirmed the lack of segregation of parasites among foxes, coyotes and dogs [59], so transmission occurs because of proximity between the wild and domestic life cycle. In addition, wolves are susceptible to the same infectious and parasitic diseases [60] as the other domestic and wild canids. We would like to highlight the need to clarify the role played by the Iberian wolf in these parasitoses and to identify the risks they could pose for animal and human health. 

## Figures and Tables

**Figure 1 animals-12-02289-f001:**
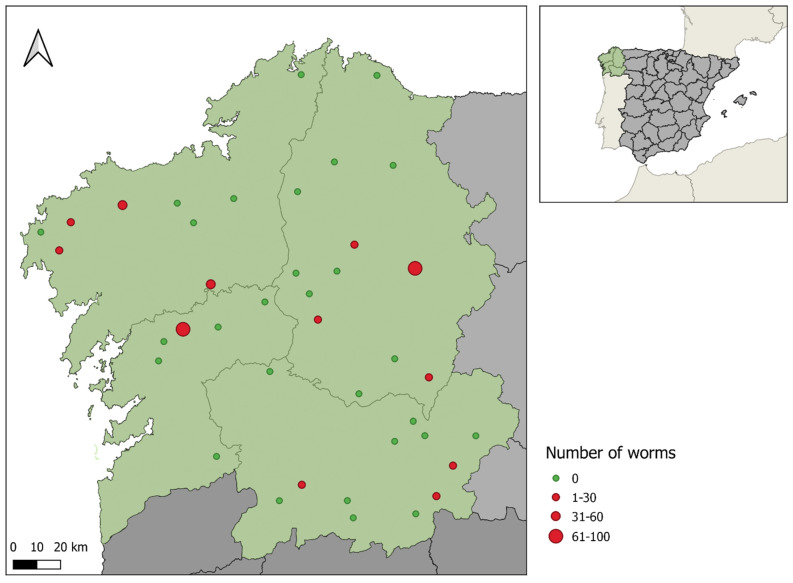
Cardiopulmonary nematode worms detected in wolves inhabiting the Galicia region (NW Spain).

**Table 1 animals-12-02289-t001:** Numbers of cardiorespiratory nematodes collected from 14 infected Iberian wolves from northwestern Spain according to the species identified.

Species Identified	No. Positive Wolves (%)	No. Total Worms	Female Worms	Male Worms	Mean (SD)	No. Worms *per* Wolf (min.–max.)
*A. vasorum*	11 (19.2)	374 *	265	71	34 (35.1)	1–99
*C. vulpis*	4 (7)	8	7	1	2 (0.8)	1–3
*E. aerophilus*	2 (3.5)	3	3	0	1.5 (0.7)	1–2
*A. vasorum* single infection	8 (14)	348 *	245	65	43.5 (36.7)	7–99
*C. vulpis* single infection	3 (5.3)	6	5	1	2 (1.0)	1–3
*A. vasorum + C. vulpis*	1 (1.7)	18	13	5	18 (0.0)	18
*A. vasorum + E. aerophilus*	2 (3.5)	13	12	1	6.5 (6.4)	2–11

* The sex of 38 worms of *A. vasorum* could not be determined: No. = number; Min. = minimum; Max. = Maximum.

**Table 2 animals-12-02289-t002:** Prevalence of *Angiostrongylus vasorum*, *Crenosoma vulpis* and *Eucoleus aerophilus* according to epidemiological category.

Category		Total No. Wolves	*Angiostrongylus vasorum*			*Crenosoma vulpis*			*Eucoleus aerophilus*			Total No. of Worms		
			No. Positive Wolves	%	*p*-Value	No. Positive Wolves	%	*p*-Value	No. Positive Wolves	%	*p*-Value	No. Positive Wolves	%	*p*-Value
Province	A Coruna	15	3	20	0.959	2	13.3	0.265	0	0.0	1	4	26.7	1
	Lugo	21	5	23.8		0	0.0		1	4.8		5	23.8	
	Ourense	15	2	13.3		2	13.3		1	6.7		4	26.7	
	Pontevedra	6	1	16.7		0	0.0		0	0		1	16.7	
Sex	Male	29	5	17.2	0.747	2	6.9	1	0	0.0	0.236	7	24.1	1
	Female	28	6	21.4		2	7.1		2	7.1		7	25.0	
Body condition (1–5)	1	2	0	0.0	0.221	0	0.0	0.444	0	0.0	1	0	0.0	0.174
	2	10	4	40.0		2	20.0		0	0.0		5	50.0	
	3	20	5	25.0		1	5.0		1	5.0		6	30.0	
	4	23	2	8.7		1	4.3		1	4.3		3	13.0	
	5	1	0	0.0		0	0.0		0	0.0		0	0.0	
Age	Pup	14	3	21.4	1	4	28.6	0.002	1	7.1	0.714	6	42.9	0.243
	Young	24	5	20.8		0	0.0		1	4.1		5	28.8	
	Adult	19	3	15.8		0	0.0		0	0.0		3	15.8	

**Table 3 animals-12-02289-t003:** Number of cardiorespiratory worms found *per* wolf according to host age.

	Wolf Pup (*n* = 14)			Young (*n* = 24)			Adults (*n* = 19)		
No. of Parasites	No.	%	*p*-Value	No.	%	*p*-Value	No.	%	*p*-Value
0	8	57.1	0.083	19	79.2	0.757	16	84.2	0.343
1–29	6	42.9	0.004	2	8.3	0.277	1	5.26	0.246
30-60	0	0	0.567	2	8.3	0.566	1	5.26	1.000
>60	0	0	1.000	1	4.2	1.000	1	5.26	1.000

No. = number.

## Data Availability

The data presented in this study are available on request from the corresponding author.

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
