# Peer review of "A High Prevalence of Cardiopulmonary Worms Detected in the Iberian Wolf (Canis lupus): A Threat for Wild and Domestic Canids"

_animals, 2022, doi:10.3390/ani12172289_

Round 1

Reviewer 1 Report

The One Health concept has given the importance it deserves to environmental and wildlife health. However, most of One Health studies related to wildlife focus on viral or bacterial zoonoses or antimicrobial resistance. This study is relevant to understand the potencial role of iberian wolves as carriers or even reservoirs of zoonotic parasites. 

The article is very well written, with good grammar, clear and concise.

The introduction is very complete, with numerous references of great relevance in the field, many of them current. Just one detail:

In line 63 you have to change "promote" to "promoting".

The methodology is very well structured and figures help understanding. Suggestions:

Line 140: "All worms collected during this procedure were deposited in tubes containing 70% ethanol"... For how long?

Line 142: There is a double space before "Worms".

Line 149: "per" should be in italics as it is a Latin word. Also modify in table 1 and lines 174 and 185.

Line 154: please add the statistical software used (commercial company and version).

The results are described in a very simple and easy-to-read way. Just a couple of suggestions:

Table 1: change "+ve" to positive.

Figures 2 and 3, I think they are interchanged.

Figure of the lung: it would be interesting to point out the parasite so that it can be found more quickly.

Line 212: There is a double space before "However".

The discussion is very good. Only a couple of things would need to be changed:

Lines 239, 260, and 292 are each double-spaced in front of "Eurasian badger", "It is today", and "this infection" respectively.

Lines 261 and 264: foci and focus should be in italics since they are Latin words.

For the rest, I think it is a very interesting study for the scientific community. Congratulations!

Author Response

The authors thank reviewer 1 for his/her useful comments, which we feel have certainly helped to improve the description of our work. The changes made to the revised manuscript have been highlighted.

Note: the line numbers are no longer the same since the order of the whole text has been modified.

Attached we provide our replies in a word file with these comments in blue.

Reviewer 2 Report

Authors give an update about the frequency of cardiopulmonary nematodes in wolves found dead from northwestern Spain through the macroscopic identification of adult worms isolated from lungs and blood vassels. Three species were detected: Angiostrongylus vasorum, Crenosoma vulpis and Eucoleus aerophilus. Reading the results, it seems that wolf can act as suitable reservoir exclusively for A. vasorum in the area. 

Comments:

Title: The term “prevalence” should be removed from the title and throughout the entire manuscript, replaced with “frequency”.

Keywords: Type the name of the species in Italics

Introduction:

Lines 56-58: mitigate this sentence “especially the latter”. Domestic ungulates are not always the main protein source that wolves use. Pezzo et al., 2003 describe that domestic ungulates represent 1/3 of the wolves’ diet in Central Italy based on the stomach and intestine analysis (Acta Theriologica 48:265-270). Figueiredo et al., 2020 state that in the last decades the wolf’s diet is shifted from domestic to wild ungulates (PLoS One 15(3): e0230433.) in a study from Portugal.

M&M:

Figure 1 is useless and should be removed. Ref (26, 27) are enough.

It is not clear what identification keys have been used, as ref [16, 28-30] perhaps are in the wrong position.

Statistical analysis: More accurate Fisher’s exact p-value should be used as the expected numbers are small. Parasite intensity and species richness do not need computation, so it is maybe better to state: “…parasite intensity and species richness were reported…”.

Results:

line 157: “The overall proportion of wolves detected positive was…”

Table 1: a little confused

The paragraph “Infection intensity” should be revised and/or re-arranged along with the one above

Table 2: Not clear what all p-values referred to. Category are wrong. Better 0; 1-29; 30-60; >60. Not clear how the variables were compared to each other

Table 3 should be in the main file

Figure 3: useless

Line 212: associated not correlated

Discussion.

Limits are missed

The role of wolves in the maintaining these parasites in the environment should be better discussed and the differences among the three species.

Author Response

The authors thank reviewer 2 for your useful comments, which we feel have certainly helped to improve the description of our work. The changes made to the revised manuscript have been highlighted.

Note: the line numbers are no longer the same since the order of the whole text has been modified.

Attached we provide the file with our replies to these comments in blue.

Reviewer 3 Report

This is a well written manuscript. The study add data on the knowledge of cardiopulmonary worms’ diversity and frequency of their detection in wolf, constituting a threat for wild and domestic canids. I support the study possible publication after appropriate minor modifications as outlined below:

In the author list please indicate the corresponding author

Line 41-42: must be “A significant relationship (p=0.001)

Line 45: “not significant” – please indicate the p value higher than 0.05

Line 125: “age). . Body” – please delete a dot

Lines 115-116: “These wolves had died of natural causes ... and were collected by official organisms ...” – please be more specific and clarify the sampling procedure and circumstances. I am note sure that all of the wolves found died were suitable for necropsy.

Lines 158-159: to be consistent with “F. hirthi”, for the mentioned species (e.g. Angyostrongylus, Crenosoma, etc.) please use the appropriate abbreviation form

Line 187: “5.26%had” – space delimitation

Line 205: please clearly indicate de presence of worms on the lung

Line 206: “nearly similar” instead of “similar”

Line 304: within the conclusion section, the authors must highlight the study limitations. Also, I would like to suggest to authors to focus only on the derived main conclusions, and shorten substantially the lines 307-313 (they seems to be speculative)

Line 316: the authors contributions are not defined in agreement with the MDPI journal indications. Please revise this concern!

Line 337: “Canis Lupus Signatus” – please italicize all of the scientific name of species throughout the reference list. Please carefully revise this concern because was completely omitted!

Author Response

The authors thank reviewer 3 for your useful comments, which we feel have certainly helped to improve the description of our work. The changes made to the revised manuscript have been highlighted.

Note: the line numbers are no longer the same since the order of the whole text has been modified.

Attached we provide our replies to your comments in blue.

Round 2

Reviewer 2 Report

Thank you. Author have rearranged and rephrased following the suggestions. In my opinion the paper is ok for the publication. 

All the best for the next research paper.